# Plants Specifically Modulate the Microbiome of Root-Lesion Nematodes in the Rhizosphere, Affecting Their Fitness

**DOI:** 10.3390/microorganisms9040679

**Published:** 2021-03-25

**Authors:** Ahmed Elhady, Olivera Topalović, Holger Heuer

**Affiliations:** 1Institute for Epidemiology and Pathogen Diagnostics, Julius Kühn Institute (JKI)–Federal Research Centre for Cultivated Plants, 38104 Braunschweig, Germany; otopalovic@agro.au.dk (O.T.); holger.heuer@julius-kuehn.de (H.H.); 2Department of Plant Protection, Faculty of Agriculture, Benha University, Moshtohor 13736, Egypt

**Keywords:** root-lesion nematode, suppressive soil, antagonistic microbes, rhizosphere, cuticle

## Abstract

Plant-parasitic nematodes are a major constraint on agricultural production. They significantly impede crop yield. To complete their parasitism, they need to locate, disguise, and interact with plant signals exuded in the rhizosphere of the host plant. A specific subset of the soil microbiome can attach to the surface of nematodes in a specific manner. We hypothesized that host plants recruit species of microbes as helpers against attacking nematode species, and that these helpers differ among plant species. We investigated to what extend the attached microbial species are determined by plant species, their root exudates, and how these microbes affect nematodes. We conditioned the soil microbiome in the rhizosphere of different plant species, then employed culture-independent and culture-dependent methods to study microbial attachment to the cuticle of the phytonematode *Pratylenchus penetrans*. Community fingerprints of nematode-attached fungi and bacteria showed that the plant species govern the microbiome associated with the nematode cuticle. Bacteria isolated from the cuticle belonged to Actinobacteria, Alphaproteobacteria, Betaproteobacteria, Gammaproteobacteria, Sphingobacteria, and Firmicutes. The isolates *Microbacterium* sp. i.14, *Lysobacter capsici* i.17, and *Alcaligenes* sp. i.37 showed the highest attachment rates to the cuticle. The isolates *Bacillus cereus* i.24 and *L. capsici* i.17 significantly antagonized *P. penetrans* after attachment. Significantly more bacteria attached to *P. penetrans* in microbiome suspensions from bulk soil or oat rhizosphere compared to Ethiopian mustard rhizosphere. However, the latter caused a better suppression of the nematode. Conditioning the cuticle of *P. penetrans* with root exudates significantly decreased the number of *Microbacterium* sp. i.14 attaching to the cuticle, suggesting induced changes of the cuticle structure. These findings will lead to a more knowledge-driven exploitation of microbial antagonists of plant-parasitic nematodes for plant protection.

## 1. Introduction

The root-lesion nematodes (RLN) from the genus *Pratylenchus* are migratory endo-parasites feeding on a vast number of economical crops. RLN are among the most damaging plant-parasitic nematodes worldwide [1]. Depending on the *Pratylenchus* species, the life cycle lasts from three to nine weeks, with all life stages being migratory and infective. RLN reside inside roots or in the soil. Inside the root, they move intracellularly causing direct damage by lesions and feeding. This leads to wilting, yellowing, necrosis, and increased susceptibility to secondary diseases [2]. While searching for a suitable host or surviving adverse conditions, these nematodes migrate through soil where they are exposed to a great variety of microorganisms. Depending on the nature of plant–nematode-microbe interactions in soil, plants may be protected by specific mutualistic microbes [3,4,5]. The term “holobiont” defines a macroorganism (plant or animal) as a unit that includes all its associated (micro)organisms [6,7]. The functioning of a macroorganism highly depends on its core microbiota [8]. As for the plant holobiont, it is still difficult to discern the role of the plant per se and the surrounding soil in shaping the microbial community that is selectively associated to the plant root. It has been suggested that the influence of roots on microbial diversity in soil varies across different soil compartments, being lower in diversity in the rhizosphere than in bulk soil [9,10]. The effects of the different plant host rhizospheres and the bulk soil microbiome against phytonematode attack were recently tested, and it was concluded that the rhizospheric microbes had a higher potential to protect plants from *Pratylenchus penetrans* and *Meloidogyne incognita* than the microbes residing in the bulk soil [11]. Furthermore, it was shown that in the case of tomato and soybean plants, the microbiome of the plant was more efficient than bulk soil in reducing nematode invasion into the roots [11], suggesting a well-established role of beneficial microbes in the plant holobiont [8]. Soil type plays an important role in shaping the diversity and richness of bacterial communities in soil [12], but the activity and the biomass of soil microorganisms in the rhizosphere is determined by root exudations [13]. Root exudates represent ions, enzymes, mucilage, and other low and high molecular weight molecules [14]. They may directly affect the microbe–nematode interactions in soil and be involved in plant defense against certain pathogens and parasites [15].

There is a specific microbial attachment to phytonematodes in the soil that is dependent on the nematode species and the soil type [16,17]. It was suggested that, besides directly antagonizing their nematode carriers, nematode-attached microbes can trigger plant defense responses against the attack of a plant-parasitic nematode [3]. The surface coat (SC) is the outermost glycoprotein layer that covers the nematode cuticle [17]. The SC is directly involved in the microbial attachment [18]. It was shown that nematodes change SC composition in response to environmental changes, which can lead to their protection against antagonizing microbes [19,20].

In the current study, we aimed to investigate how the factors that contribute to the establishment of a plant holobiont, also contribute to the establishment of a nematode holobiont. We hypothesized that nematodes engage in a prolonged dialogue with their host plants directly prior to the invasion at the soil–root interface, where roots are always enriched with specific taxa of microorganisms and different molecules. This determines the microbiome associated with their bodies, which shifts among different host plants and affects their fitness. To test our hypothesis, we employed culture-independent and culture-dependent methods to study the microbial attachment to *P. penetrans* in two different soil types, and compared the attachment preferences of microbiomes residing in the rhizospheres of maize, soybean, Ethiopian mustard, oat, and tomato plants to the attachment of the microbiome from bulk soil. Nematode-attached bacteria were isolated and identified. In addition, the effects of the attached soil microbiome and single bacterial isolates were tested against nematode mortality and motility in vitro. As it was shown that the root exudates of different plants affect the attachment of *Pasteuria* spores to root-knot nematodes [21,22], and can directly affect changes in nematode gene expression [23,24], we also studied whether the root exudates affect changes in the attachment of soil microbes or single bacterial isolates to *P. penetrans*.

## 2. Materials and Methods

### 2.1. Preparation of Nematodes

Fresh mixed stages of *P. penetrans* were extracted from a 2-month axenic carrot disc culture using a Baermann funnel and surface-sterilized according to Elhady et al. [11]. Briefly, nematodes were washed over 5-µm sieves with 10 mL of sterilized tap water and incubated in 0.02% HgCl_2_ for 3 min, followed by incubation in 5 mL 1 × CellCultureGuard (AppliChem, Darmstadt, Germany) for four hours. Nematodes were recovered on sterile 5-µm sieves (Cell-Trics1 filters, Sysmex, Norderstedt, Germany), washed with 10 mL of sterile tap water, and incubated for 48 h in sterile tap water to renew their cuticle, followed by another washing step over sterile 5-µm sieves to wash out the debris of the old cuticle directly before the use in experiments.

### 2.2. Experimental Design: Microbiome Associated with RLN as Affected by Plant Species

We tested whether the plant species governs the microbiome associated with the RLN cuticle in two independent experiments.

#### 2.2.1. Preparation of Rhizosphere and Bulk Soil Suspensions

In the first experiment, the rhizosphere microbiome was obtained from different plant hosts, maize (*Zea mays* L. cv. Colisee), soybean (*Glycine max* L. cv. Primus), and tomato (*Solanum lycopersicum* L. cv. Moneymaker). The plants were grown in 12-cm diameter plastic pots filled with 500 mL of field soil (less sandy loam with 1.4% humus, pH 6.2; 52°17′57″ N, 10°26′14″ E). The bulk soil was obtained in the same way, without growing a host plant. In the second experiment, soil suspensions from the rhizospheres of maize, Ethiopian mustard (*Brassica carinata* cv. Cappuccino), and oat (*Avena strigosa* cv. Luxurial) plants grown in field soil (sandy loam, pH 6.3; 52°16′21.7″ N 10°34′02.7″ E) were obtained in the same way to study the attachment of bacteria and fungi to *P. penetrans* in this soil. The pots of both experiments were irrigated and fertilized using 2.5 g L^−1^ of a commercial fertilizer (WUXAL Super NPK fertilizer, 8-8-6 with micronutrients, AGLUKON, Düsseldorf, Germany), and kept for 6 weeks in the greenhouse at 24 °C and with a 16:8 h photoperiod. To extract the microbiome, 5 g of roots with attached soil or bulk soil was blended in a Stomacher blender (Seward, London, UK) with 15 mL of sterile 0.85% NaCl at high speed for 60 s to efficiently release the microbes into the suspension. The supernatants were sieved through 5-µm sieves to remove mesofauna, root debris, and remaining soil particles.

#### 2.2.2. Baiting of Soil Microbes Attaching to the Cuticle of *P. penetrans*

Baiting of microbes from the rhizosphere or bulk soil suspensions on the cuticle of *P. penetrans* was done as previously described [25]. Briefly, 20,000 surface-sterilized nematodes were incubated in 5 mL of each microbial suspension in 15 mL tubes. The tubes were positioned horizontally on a shaker at a speed of 150 rpm at 20 ± 2 °C for 24 h. The nematodes with attached microbes were recovered on 5-µm sieves and washed with 20 mL of sterile tap water to remove loosely attached microbes. To isolate the attached bacterial strains, nematodes were plated on R2A media (Merck, Germany) supplemented with 10 mg L^−1^ cycloheximide. The plates were incubated at 28 °C and bacterial strains were collected from the emerged colonies over a 2-week period. A portion of the nematodes with attached microbes was transferred to bead-beating tubes with Lysing Matrix E (MP Bio, Heidelberg, Germany) for DNA extraction.

#### 2.2.3. Denaturing Gradient Gelelectrophoresis (DGGE) Profiling of Nematode-Attached Microbiomes

Total DNA from *P. penetrans* and their associated microbes was extracted using the Fastprep FP120 bead beating system for 30 s at a high speed and a FastDNA Spin Kit for Soil (MP Bio, Heidelberg, Germany). The DGGE profiling of the nematode-attached microbiome was done according to Adam et al. [16]. The bacterial 16S rRNA fragments from low DNA concentrations were amplified by nested-PCR, using in the first PCR the primer pair S-D-Bact-0008-a-S-16/S-D-Bact-1492-a-A-16, followed by a second PCR using the primer pair F984GC/R1378 [26]. The fungal internal transcribed spacers (ITS) were amplified in a nested PCR approach using the primer pairs ITS1F/ITS4 and ITS1FGC/ITS2 [27]. DGGE was done using the PhorU2 system (Ingeny, Goes, The Netherlands) according to Weinert et al. [27]. The software GelCompar II v. 6.6 (Applied Maths, Sint-Martens-Latem, Belgium) was used to analyze the DGGE profiles. The resulting Pearson similarity matrices were used for a permutation test on significant differences among the microbial communities [28]. Some of the bacterial and fungal bands were identified by the cloning of PCR products using the vector pGEM-T and *Escherichia coli* JM109 high-efficiency competent cells (Promega, Madison, WI, USA), and sequencing with vector primers T7 and SP6 (Macrogen, Amsterdam, The Netherlands). The sequences of fungal and bacterial clones, as well as isolates, were deposited in NCBI GenBank with accession numbers MN332046 to MN332063, and MW326933 to MW326970.

#### 2.2.4. Characterization of Bacterial Isolates

Morphologically different nematode-attached bacterial isolates were purified twice to obtain pure isolates. To isolate the DNA from bacterial isolates, bacterial cells were lysed by adding 100 µL of 50 mM Tris-HCl pH 8.0/50 mM EDTA/0.5% Tween 20/0.5% Triton X-100, containing 200 µg lysozyme, 90 µg proteinase K, and 20 µg RNase A. After a 30 min-incubation at 37 °C, 3 M guanidine hydrochloride/20% tween 20 was added, and the lysate was incubated at 50 °C for 30 min. A 200 µL suspension of GeneClean Spin Glassmilk (MP Bio) was added to capture DNA from the lysate. The pelleted Glassmilk was washed twice with 500 µL washing solution (100 mM NaCl/1 mM EDTA/10 mM Tris-HCl, pH 7.5/50% EtOH). The DNA was air-dried for 10 min and eluted with 100 µL 10 mM Tris-HCl/0.1 mM EDTA pH 8.0. The supernatant containing DNA was separated from the Glassmilk by centrifugation at a maximum speed for 2 min (12,000× *g*) and stored at −20 °C until use. Discrimination of isolates to identify unique strains was performed by BOX-PCR fingerprinting [29]. From ca. 20 ng of template DNA, genomic fragments were amplified in a 25 µL PCR reaction using GoTaq Flexi Buffer, 3.75 mM MgCl2, 0.2 mM of each dNTP, 5% *w/v* DMSO, 0.2 µM primer BOXA1R (5′-CTA CGG CAA GGC GAC GCT GAC TGA CG-3′), and 1 U GoTaq Flexi DNA polymerase (Promega). PCR conditions were as follows: denaturation step for 7 min at 94 °C, 35 cycles of 1 min at 94 °C, 1 min at 53 °C, and 8 min at 65 °C, and a final extension step for 16 min at 65 °C. The PCR bands were separated by 1.5% agarose gel electrophoresis for 3 h at 80 V and visualized by UV transillumination (254 nm) after staining with ethidium bromide. The band patterns of the bacterial isolates were compared by GelCompar II 6.6 (Applied Maths, Sint-Martens-Latem, Belgium), and twenty different isolates were selected for further investigations.

### 2.3. Biological Effects of the Nematode-Attached Microbiome

To study whether the microbiome that attaches to *P. penetrans* affects nematode viability, soil suspensions were prepared from bulk soil and the rhizospheres of Ethiopian mustard, maize, and oat. For preparation of soil suspensions, 10 g of soil or roots with adhering rhizosphere soil were blended using a Stomacher 80 blender (Seward) in 2 × 20 mL of sterile tap water for 1 min at a high speed. Soil particles were spun down for 5 min at 500× *g* and the supernatant was sieved through a sterile 5 µm sieve (Cell-Trics1 filters, Sysmex, Norderstedt, Germany). The flow-through was pelleted for 10 min at 5000× *g* and the pellet re-suspended in 6 mL of sterile tap water. Around 4000 surface-sterilized nematodes were incubated in 4 mL of soil suspensions or sterile tap water as a control at low shaking speed overnight. The next day, nematodes were recovered on 5-µm sieves and washed with 15 mL sterile tap water to remove loosely attached microbes. A subset of 100 washed nematodes, with or without attached microbiome, was incubated for 5 days in sterile tap water to assess nematode viability. To determine the number of attached bacteria per worm, nematodes were plated on R2A media supplemented with 10 mg/L cycloheximide. The R2A plates were kept at 28 °C for 2 days before counting colony-forming units (CFU).

To study the effects of single bacterial isolates on nematode mortality, bacterial cultures of isolates i1- i55 (Table 1) were grown from 100 µL pre-culture in 25 mL LB broth (Luria/Miller-Carl Roth, Karlsruhe, Germany) for 24 h at 28 °C with shaking. Strains *Rhizobium etli* G12 and *E. coli* JM109 were used as positive and negative controls, respectively. Bacterial cultures were centrifuged at 4000× *g* for 15 min to obtain the bacterial cells. Later, pellets of bacterial cells were re-suspended in 1 mL 0.85% NaCl and the concentration was adjusted to 0.2 OD_600nm_. A volume of 50 µL sterile suspension comprising 500 nematodes was added to 2 mL of bacterial suspensions in 24-well plates (Carl Roth, Karlsruhe, Germany). The 24-well plates were incubated at 20 ± 2 °C with slow shaking. Live and dead nematodes were evaluated using a stereomicroscope (Olympus Microscope SZX12) after 48 h.

### 2.4. Effects of Root Exudates on Microbial Attachment to Nematodes 

To obtain root exudates, the seeds of maize, soybean, and tomato were surface sterilized using 1.5% sodium hypochlorite for 15 min and rinsed five times with sterile deionized water. The seeds were planted in sterile jars containing 1/2 strength MS media (Murashige and Skoog medium including vitamins, DUCHEFA BIOCHEMIE, Netherlands) and maintained in a growth chamber for 2 weeks at 22 °C (65% humidity and 16 h photoperiod). To collect the root exudates, plants were carefully removed from the growth substrate. The roots were gently washed and incubated in 50 mL of sterile water in glass jars. After 48 h, the released root exudates were collected, filter-sterilized through a 0.2-µm filter (Minisart, Sartorius Stedim biotech, Göttingen, Germany) and stored at −20 °C until use.

To test the effects of root exudates on microbial attachment to *P. penetrans*, around 5000 surface-sterilized nematodes were incubated in 2 mL of root exudates of soybean, maize, tomato, or 1 µM α-naphthalene acetic acid (NAA, Duchefa Biochemie, Netherlands) and sterile tap water as control in 6-well plates (Carl Roth, Karlsruhe, Germany) at 20 ± 2 °C overnight with slow shaking. The following day, 2 mL of the bulk soil suspension was added to each well and the incubation continued overnight. After incubation, nematodes were recovered on 5-µm sieves and washed with 15 mL of sterile tap water to remove loosely attached microbes. Nematodes with attached microbes were transferred to bead-beating tubes and stored at −20 °C until DNA extraction.

To test if root exudate-induced changes of the nematode cuticle affect attachment of bacterial isolates, 2 mL microtubes containing 500 nematodes in 500 µL of soybean, maize, or tomato root exudates were incubated at 20 ± 2 °C with slow shaking overnight. Incubations in sterile tap water and 1 µM NAA solution served as controls. The following day, nematodes were spun down in 1.5 mL microtubes at 1000× *g* for 2 min, and the supernatant was replaced with 250 µL of the bacterial strain i.14 and 250 µL of the bacterial strain i.14-Rif as a reference. Nematodes were additionally incubated in bacterial suspensions overnight. Non-attached bacterial cells were separated from the nematodes by centrifugation at 1000× *g* for 1 min at room temperature. The supernatant was discarded and the nematode pellet was washed twice by centrifugation with 1 mL of sterile tap water. Nematodes with attached bacterial cells were re-suspended in 1 mL of sterile tap water by adding 0.2 g of 0.1 mm glass beads and vortexing for 10 s. To determine the number of attached bacterial CFU per nematode, these were serially diluted and plated on R2A media. The plates were kept at 28 °C and the CFU were counted after 48 h.

### 2.5. Data Analysis and Statistics

The GENMOD procedure from the package SAS 9.4 (SAS Institute Inc., Cary, NC, USA) was used to analyze count data. It was performed with Poisson distribution, log link function, and specification of a scale parameter (Pearson) to account for overdispersed data. For multiple comparisons to a control, the alpha level was adjusted according to Dunnett. For multiple comparisons, the Tukey’s significant post hoc test was used to obtain significant differences represented in letters. For the non-metric multidimensional scaling (NMDS), Pearson correlations of background-subtracted densitometric curves from the DGGE analysis were processed using the R package Vegan. The DGGE community profiles were compared using software GelCompar II 6.6 (Applied Maths, Sint-Martens-Latem, Belgium). Pearson correlation for calculating similarity coefficient values per lane was used for clustering based on the unweighted pair group method with arithmetic mean (UPGMA) and to perform permutation tests for significant differences among nematode associated microbes from the different rhizosphere and bulk soils, and to calculate the d-value that indicates the difference in average similarities within and among groups according to Kropf et al. [28].

## 3. Results

### 3.1. Rhizosphere Fungi Attaching to P. penetrans Depended on Plant Species

PCR-DGGE fingerprinting of fungal ITS and bacterial 16S rRNA gene fragments was used to compare the microbial communities that specifically attached to *P. penetrans* in the rhizosphere of different plant species or in bulk soil. Two independent experiments were performed, with tomato, maize, and soybean in the first experiment, and with maize, oat, and Ethiopian mustard in the second experiment. The fungal species attaching to *P. penetrans* differed significantly among the rhizospheres of the plant species (*p* < 0.001, *n* = 4, constrained permutation test based on pairwise Pearson correlations of the PCR-DGGE fingerprints; Figure 1). The cuticle-attached fungal communities in the rhizosphere differed among plants by 43–87% in the first experiment, and 3–29% in the second experiment (Table 2). The fungal communities attaching to *P. penetrans* cuticle from maize and tomato rhizosphere were most dissimilar, while in the rhizosphere of oat and Ethiopian mustard the fungal communities attaching to the cuticle were highly similar. Moreover, the results revealed a significant difference of the cuticle-attached fungal communities between rhizospheres and bulk soil (*p* < 0.001), however, these communities from rhizospheres differed more from those from bulk soil than among rhizospheres of the different plants (Table 2). The dissimilarity of nematode-associated fungal communities between bulk soil and rhizospheres was 62% to 88%, and 10% to 28% in the first and second experiment, respectively (Table 2). The abundant soil fungi that were not attached to the cuticle significantly differed among rhizospheres and between bulk soil and rhizosphere soils (*p* < 0.001, Appendix A). The abundant fungi in soil were more diverse than cuticle-attached fungi (Appendix A compared to Figure 1), and the fungal communities in soil were distinct from those on the nematodes, as shown by NMDS (Figure 2).

In the first experiment, the fungal fingerprints showed 14 distinct bands (Figure 1). One fungal type was associated to the nematodes in all the treatments (band no. 11 in Figure 1). The DNA of the band was sequenced and assigned to *Cladosporium tenuissium* with 99% identity (Appendix A). This fungus was the most abundant on the cuticle after incubation of the nematodes in soybean and tomato rhizosphere soil suspensions (Figure 1). The band was also detectable in the nematode inoculum after surface-sterilization, but with a very low intensity. The fungal type identified as *Malassezia restricta* with 99% identity (band no. 1, Figure 1) attached to *P. penetrans* in the rhizosphere soil suspensions of all three plants, but not in the bulk soil suspension. Although several nematode-attached fungal types were shared between nematodes from two different treatments (bands no. 2, 6, and 12; Figure 1 and Appendix A), most attached fungi exhibited a high specificity for one of the rhizospheres. For instance, the fungal bands assigned to *Myrothecium verrucaria* with 99% identity (band no. 7) and *Aspergillus tonophilum* with 100% (band no. 10) identity were associated with *P. penetrans* only after incubation in the suspension of the maize rhizosphere. Other fungi attached to the nematodes only in the suspension of tomato rhizosphere soil: band no. 8, identified as *Penicillium alli*, *P. gladioli*, and *P. hordei*; band no. 13 identified as *Sporidiobolus pararoseus*; and band no. 14 assigned to *Cutaneotrichosporon curvatus* (Appendix A).

### 3.2. Rhizosphere Bacteria Attaching to P. penetrans Depended on Plant Species

Similarly to what we observed for fungi, the bacterial attachment to *P. penetrans* in rhizosphere soil suspensions was plant-specific in the two experiments (*p* = 0.032 for experiment 1; *p* = 0.018 for experiment 2). The bacterial DGGE profiles comprised 19 bands of cuticle-attached bacteria in the rhizosphere and 12 bands in bulk soil (Figure 3). Three bacterial types were commonly associated with *P. penetrans* in bulk soil and in the maize and soybean rhizospheres (Figure 3): band no. 6, assigned to *Bradyrhizobium embrapense* with 95% identity; band no. 8, assigned to *Pseudomonas synxantha*; and band no. 12, assigned to *Streptococcus himalayensis* (Appendix A). However, few other bands were common among treatments supporting a very specific association between nematodes and bacteria in each rhizosphere soil: *Streptococcus rubneri* (band 9) and *Moraxella nonliquefaciens* (band 13) were exclusively detected on the nematode’s cuticle after incubation in the suspension of tomato rhizosphere soil; *Paraburkholderia dipogenis* (band 1) was specifically associated with nematodes in soybean rhizosphere suspension.

The abundant bacteria in soil were much more diverse than cuticle-attached bacteria (Appendix A compared to Figure 3). The bacterial communities in soil were distinct from those attached to the nematodes, as shown by NMDS (Figure 4). The cuticle-attached bacterial communities in the rhizosphere differed among plants by 43–68% in the first experiment, and 22–36% in the second experiment (Table 2). The community attached to *P. penetrans* from soybean and tomato rhizosphere were most dissimilar, while in the rhizosphere of oat and Ethiopian mustard the bacterial communities attaching to the cuticle were most similar. It also differed between rhizospheres and bulk soil (*p* < 0.001). The dissimilarity of nematode associated bacterial communities between bulk soil and rhizospheres was 23% to 40%, and 12% to 34% in the first and second experiment, respectively (Table 2). The abundant soil bacteria that were not attached to the cuticle significantly differed among rhizospheres and between bulk soil and rhizosphere soils (*p* < 0.001, Appendix A). The differences among the microbial communities that attached to nematodes in the different soils and the non-attached microbial communities were not well correlated (Appendix A).

### 3.3. Isolation and Characterization of Cuticle-Attached Bacteria

The isolation of culturable bacterial strains with high affinity to attach to the cuticle of *P. penetrans* baited in different rhizospheres and corresponding bulk soil resulted in 20 isolates with unique fingerprints, as characterized by 16S rRNA gene sequencing (Table 1, Appendix A). Nine bacterial species were isolated from the nematodes after the incubation in the suspension of bulk soil, while the diversity of isolates obtained after incubation of the nematodes in the rhizosphere suspensions was lower. The bacterial isolates belonged to Actinobacteria (*Streptomyces*, *Pseudomonas*, *Microbacterium, Mycobacterium, Nocardia*), Alphaproteobacteria (*Rhizobium, Novosphingobium*), Betaproteobacteria (*Delftia, Alcaligenes*), Gammaproteobacteria (*Lysobacter*), Sphingobacteria (*Pedobacter*), and Firmicutes (*Bacillus, Staphylococcus*).

Nine bacterial strains that were isolated from the cuticle of *P. penetrans* were tested for their attachment to the nematodes (Figure 5). Among these, only two strains, i.16 and i.35, showed a low attachment rate with less than 30 attached CFU per nematode, while all the other strains had a significantly higher attachment rate to *P. penetrans* than the negative control strain *E. coli* JM109 (*p* < 0.0001, Dunnett’s test). The highest density of attached cells was observed for the bacterial isolates *Microbacterium* sp. i.14, *Lysobacter capsici* i.17, and *Alcaligenes* sp. i.37.

### 3.4. Effect of Cuticle-Attached Microbiomes on the Mortality of P. penetrans

We tested the capacity of the nematode-attached microbiomes from rhizospheres of Ethiopian mustard, maize, and oat, and from the corresponding bulk soil to affect the mortality of infective stages of *P. penetrans*. After baiting nematodes in the soil suspension, cuticle-attached microbes per nematode ranged between 26 CFU for Ethiopian mustard rhizosphere and 93 CFU for oat rhizosphere (Figure 6). The microbial attachment was significantly higher for oat rhizosphere and bulk soil compared to Ethiopian mustard rhizosphere. In the in vitro mortality assay, all attached microbiomes increased the percentage of dead nematodes compared to the control, where nematodes were incubated in sterile tap water (Figure 7). However, the numbers of dead nematodes were low, ranging from six nematodes killed by the microbiome from bulk soil to nine nematodes killed with the microbiome from the rhizosphere of Ethiopian mustard. The numbers of microorganisms attaching to nematodes in the different treatments did not correlate well with mortality.

### 3.5. Effect of Cuticle-Attached Bacterial Isolates on the Mortality of P. penetrans

To monitor the nematicidal activity of bacteria isolated from the cuticle of *P. penetrans*, nematodes were incubated in suspensions of twenty different bacterial isolates over a 48-h period. The effects on nematode survival were compared to sterile tap water as a negative control. The nematicidal effect of the bacteria was highly variable, ranging from less than 10% for four strains up to 95% mortality for *Bacillus cereus* i.24 (Figure 8). The latter had a significantly higher efficiency than the well-studied antagonistic strain *Rhizobium etli* G12 [30] (83.5% mortality on average). Isolate *L. capsici* i.17 also efficiently antagonized *P. penetrans*, leading to 79.5% mortality (Figure 8).

### 3.6. Effects of Root Exudates on the Attachment of Bacteria

Differences in the microbiome structure on the cuticle of *P. penetrans* were associated with differences in the rhizosphere microbiome of different plant species. We wanted to test whether differences in root exudates of the plant species also contribute to differential attachment of microbes by changing the surface of the nematode. Surface-sterilized *P. penetrans* were exposed to sterile filtered root exudates from maize, soybean, or tomato. Sterilized tap water or an auxin solution served as control. Subsequently, the nematodes were baited in a bulk soil suspension overnight, washed, and the attached microbiome was studied by DGGE fingerprinting (Appendix A). Overall, pre-incubation of nematodes in the different root exudates or in an auxin solution did not significantly affect the composition of bacterial or fungal species that attached to the nematode surface in soil suspension. However, some additional bands in the bacterial DGGE profiles indicated bacterial species that specifically attached to the cuticle of those nematodes, which were pretreated with maize or soybean exudates (bands marked by an arrow in Appendix A). These differences were less notable for the attached fungal communities (Appendix A).

However, baiting of the differently conditioned nematodes in a suspension of the bacterial strain *Microbacterium* sp. i.14 resulted in a significantly reduced attachment of the bacteria to the cuticle of *P. penetrans* that was treated with exudates from soybean or maize roots compared to the other treatments (Figure 9). This suggested an effect on the cuticle structure by some plant-specific components of the root exudates. Auxin at a 1 µM concentration did not induce cuticle changes. Exposure to exudates from tomato roots resulted in a trend for reduced attachment of strain i.14 that was not statistically significant compared to the water control using Tukey’s test.

## 4. Discussion

### 4.1. Plants Govern the Microbiome Associated with the Cuticle of P. penetrans in the Rhizosphere

In a previous study, the rhizosphere microbial communities of maize, but not tomato, protected plants better against nematode attack than the microbial community from bulk soil [11]. As cuticle-attached microbes can affect the root invasion of nematodes [16,25,31,32], we investigated whether shifts in soil microbial communities induced by different plant species led to a different set of microbes attaching to the phytonematode species *P. penetrans*. DGGE fingerprints of fungal ITS fragments and bacterial 16S rRNA genes showed that the plant species determined which microbes attached to the cuticle. The microbial community structure in the rhizosphere was highly dependent on the plant species. This is consistent with several studies that emphasized the importance of plant species and soil type in the community structure of rhizosphere microorganisms [8]. The portion of bacteria and fungi that attached to *P. penetrans* after incubation in the soil suspensions of bulk soil and the different rhizosphere soils was very small in comparison to the total soil microbial communities. This means that the attachment of soil microorganisms to the nematodes is very specific and selective, and relies on the soil type and the nematode species [16,30]. In addition, the structure of the nematode-attached microbiome was very different depending on the rhizosphere type. This was especially obvious for nematode-attached fungi.

Given that nematodes migrate from bulk soil targeting their host plants, several factors can contribute to the establishment of the attached microbiome, which in turn can determine the nematode’s behavior and interaction with the host plant. In the rhizosphere, microbial diversity is less than in bulk soil while specific microbial species are much more active and relatively more abundant due to the selection by root exudates [33]. Nematodes can modify their behavior on the basis of plant signals such as phytohormones that are present in root exudates. These may trigger rapid changes in the surface structure of plant-parasitic nematodes [34,35]. This kind of change could be a reflection of the regulation of cuticle encoding genes leading to the preferential attachment of specific microbial species. For instance, the attachment of endospores of the nematophagous bacterium *Pasteuria* was found to be increased in response to root exudates, depending on the plant species and the presence of other soil microorganisms [21,22]. Nematodes modify their surface coat by producing surface proteins that mimic the ones produced by the host to counteract the host plant defense and escape from host detection [36,37].

In parallel, microbes that reside in the root–soil interface have evolved different means to enable them to navigate, respond, and bind to their hosts. For instance, soil bacteria move towards their host via multiple cellular chemotaxis and chemoreceptor encoding genes, which were found to be enriched in the rhizosphere compared to bulk soil in response to gradients of compounds derived from the host plant. Several types of chemotaxis signaling in beneficial bacteria, which promote motility through flagella and pili, are prevalent in the rhizosphere [38,39]. Adhesion factors in microbes that are stimulated in response to plant signals may determine the specific binding to the cuticle of nematodes. Bacteria and fungi bind to biomaterials through a wide variety of surface proteins, like adhesins or filaments that decorate the bacterial cell with appendages. For instance, the adhesion of *Azospirillum brasilense* was found to be affected by plant species specific compounds. The major outer membrane protein (MOMP) of *A. brasilense* had a stronger adhesion to the membrane-immobilized roots of cereals than to legumes or tomato extracts [40]. In light of this, our results demonstrate that plant specific root exudates and the microbiome in the rhizosphere determine the microbiome associated with the cuticle of the nematode directly before root invasion. This in turn might affect their fitness and invasion into the root. Consequently, deciphering the molecular interaction network of plant, nematode, and microbes will eventually lead to knowledge-based control of plant-parasitic nematodes.

### 4.2. Effects of Nematode-Attached Microbiome on Nematode Fitness

Using a culture-dependent approach, we isolated bacterial strains that specifically attached to *P. penetrans* in the rhizospheres of soybean, maize, and tomato, or in bulk soil. The ability of the isolates to attach to *P. penetrans* was confirmed in re-attachment experiments where only two strains showed attachment rates as low as that of the control strain *E. coli*, which was demonstrated to have a low affinity to attach to the cuticle [41]. The most prominent strains that highly attached to *P. penetrans* were assigned to the genera *Streptomyces, Pseudomonas, Microbacterium, Mycobacterium, Rhizobium, Lysobacter, Alcaligenes*, and *Bacillus* based on 16S rRNA sequence analysis. All of these bacterial genera have been frequently reported to antagonize RLN or root-knot nematodes [42,43,44,45,46,47,48,49,50,51]. Several isolated strains showed higher nematicidal effects in vitro than the well-known positive control *R. etli G12*, including *B. subtilis* and *L. capsici*. Siddiqui and Mahmood [50] proposed that non-parasitic rhizobacteria can antagonize nematodes by the production of specific toxic metabolites and by alteration of root exudates. For instance, two toxic compounds, fervenulin and 6,8-dixydroxy-3-methylisocoumarin, that were isolated from a strain of *Streptomyces* sp., were able to antagonize *Meloidogyne incognita* in vitro by impeding egg hatch and by accelerating J2 mortality [52]. In addition, both *B. cereus* S2 and its secondary metabolite sphingosine were lethal for *Caenorhabditis elegans* and *M. incognita* in vitro or by inducing systemic resistance in plants [45]. In fact, evidence is accumulating that many rhizobacteria, including *Bacillus* spp., go as far as triggering expression of PTI-responsive defense genes in plants that hinder nematode penetration and development [32,53,54,55,56,57,58]. With that in mind, the ability of isolated strains in our study to function in nematode suppression by inducing systemic resistance in plants should be further investigated.

It has been proposed that rhizo-microbiomes show high efficiency in nematode control for two reasons: (1) their relative abundance and activity are generally higher in the rhizosphere than in the surrounding soil [9,10]; and (2), the phytonematodes reside in close proximity of the root, thus sharing the rhizosphere habitat with plant-associated microbes [59]. In our study, the nematode mortality induced by the attached microbiome from rhizospheres of Ethiopian mustard, maize and oat, and from bulk soil was rather low, and was independent from the number of attached bacterial cells on the nematode after a 24-h incubation. While higher microbial diversity and species richness in the rhizosphere would presumably lead to a higher nematode mortality [60], we think that the species evenness of microorganisms that managed to attach to *P. penetrans* was low and insufficient to cause a high mortality effect, as it was the case with single bacterial isolates. Furthermore, some studies showed that microorganisms alone did not affect nematode mortality and movements in the absence of the host plant, but they prevented nematode attraction and performance on the roots in the case where the plant was present [25,61].

### 4.3. Role of Root Exudates in Microbial Attachment and Suppression of P. penetrans

Root exudates contain molecular cues that are of importance for the cross talk between plants, microorganisms, and nematodes [15]. They were reported to directly induce changes in the nematode’s surface composition, which was measured by the nematode’s ability to absorb lipid probes after exposure to different environmental cues [17,34,35]. Potato root diffusates have shown to increase the uptake of the lipid probe AF18 by *Globodera rostochiensis* [34], while tomato root diffusates increased the surface lipophilicity of *M. incognita* [35]. Phytohormones, such as kinetin and auxin, also induced differential changes in surface lipophilicity [34]. This prompted us to investigate whether the changes in surface composition of *P. penetrans* by different root exudates would affect the binding of the bacterial isolate *Microbacterium* sp. i.14, which showed a high rate of attachment to this nematode species. Interestingly, we found that the attachment of this isolate was reduced by root exudates from soybean and maize, but not from tomato. The increased bacterial binding after exposure to tomato root exudates reflected the high permissiveness of this plant to nematode attack. Yang et al. [62] found negative effects of root exudates from both resistant and highly susceptible tomato cultivars on the egg hatch and J2 mortality of *M. incognita*, but the J2 attraction to the roots was promoted by the latter.

## 5. Conclusions

In our study, we found that the host plant plays an important role in shaping the nematode–microbiome association. Microbial taxa or species that are enriched in the rhizosphere can affect the fitness of parasitic nematodes. Each plant species recruits its associated microbial species based on the soil type and availability of diverse taxa of microorganisms that, in turn, support suppressiveness against plant-parasitic nematodes. Thus, the plant–soil feedback determined by the interactions of root exudates, nematode-attached microbes in the rhizosphere, and the nematode cuticle might be important to understand soil suppressiveness against plant-parasitic nematodes. 

## Figures and Tables

**Figure 1 microorganisms-09-00679-f001:**
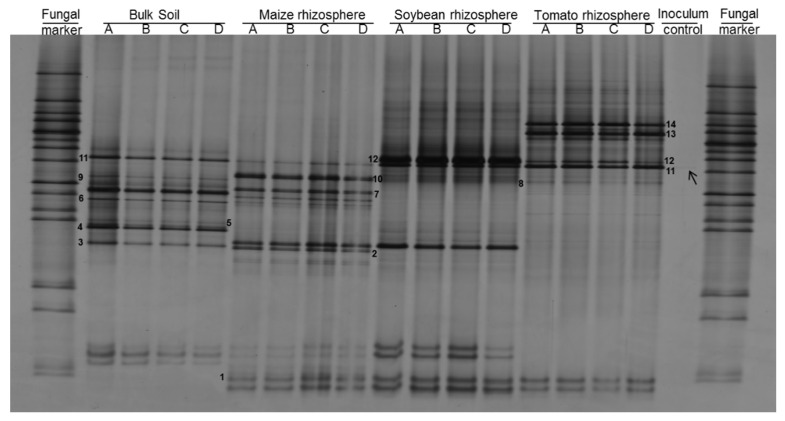
Denaturing gradient gelelectrophoresis (DGGE) profiles of fungal internal transcribed spacer (ITS) fragments amplified from DNA of *Pratylenchus penetrans* incubated in suspensions of bulk soil and rhizosphere soils (Experiment 1). Letters A, B, C, and D represent biological replicates. Surface-sterilized nematodes prior to incubation are referred to as inoculum control. Numbers 1–14 represent the bands that were taxonomically assigned by DNA sequencing and that showed specificity of fungal species in their attachment to *P. penetrans*.

**Figure 2 microorganisms-09-00679-f002:**
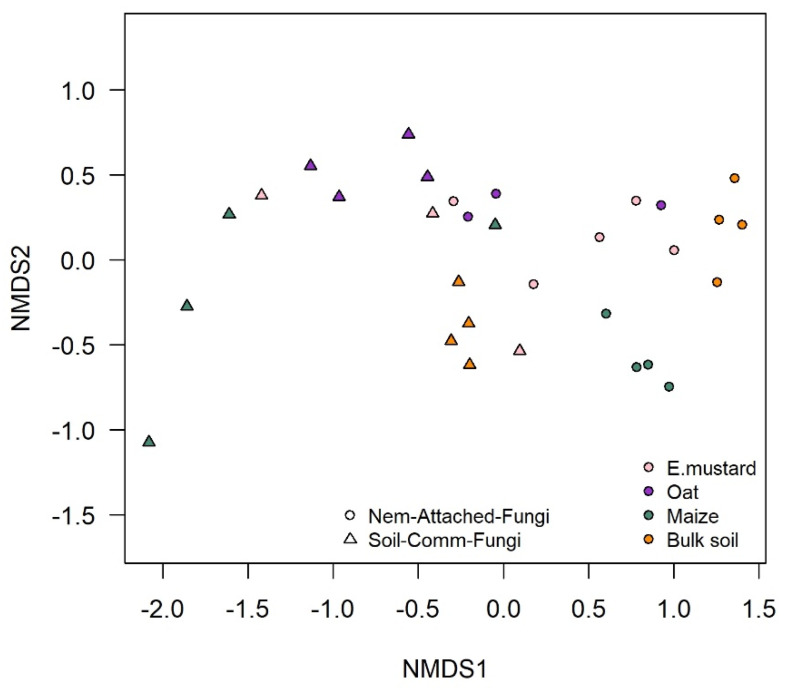
Effect of plant species on soil and *P. penetrans* attached fungi. Non-metric multidimensional scaling (NMDS) plots are presented based on Pearson correlations of background-subtracted densitometric curves from denaturing gradient gelelectrophoresis (DGGE). Stress value is 0.031, *n* = 4.

**Figure 3 microorganisms-09-00679-f003:**
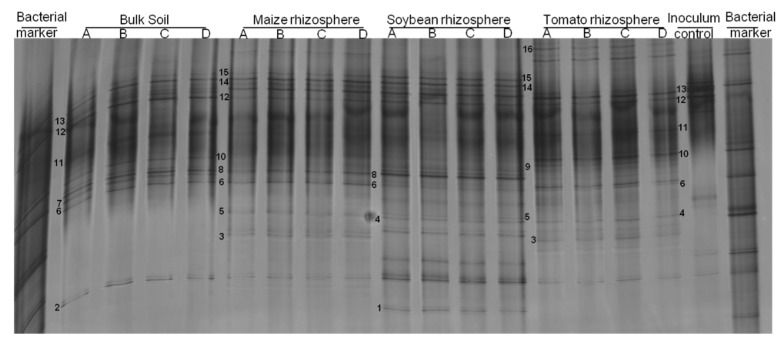
Denaturing gradient gelelectrophoresis (DGGE) profiles of bacterial 16S rRNA fragments amplified from DNA of *Pratylenchus penetrans* incubated in microbial suspensions of bulk soil and the rhizospheres of maize, soybean, and tomato plants grown in the same soil (Experiment 1). Letters A, B, C, and D represent biological replicates. Surface-sterilized nematodes prior to incubation are referred to as inoculum control. Numbers 1–15 represent bands that were taxonomically assigned by DNA sequencing and that showed specificity of bacterial species in their attachment to *P. penetrans*.

**Figure 4 microorganisms-09-00679-f004:**
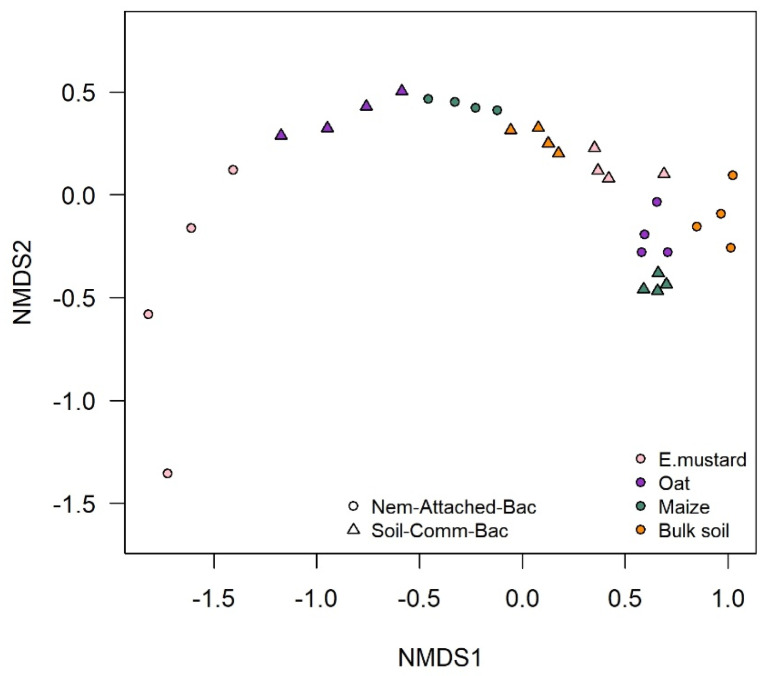
Effect of plant species on soil and *P. penetrans* attached bacterial communities. Non-metric multidimensional scaling (NMDS) plots are presented based on Pearson correlations of background-subtracted densitometric curves from denaturing gradient gelelectrophoresis. Stress value is 0.037, *n* = 4.

**Figure 5 microorganisms-09-00679-f005:**
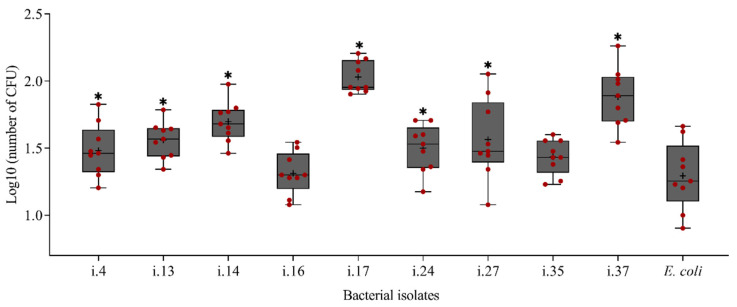
Attachment capacity of bacterial isolates from infective stages of *Pratylenchus penetrans* to the cuticle of *P. penetrans*. Nematodes were incubated in suspensions of the bacteria in water, washed on sterile sieves, and plated on R2A media. *Escherichia coli* JM109 was used as a negative control. Mean log transformed numbers of CFU are shown as (+) for each strain, the medians are shown as (—), whiskers indicate quartiles. Stars above whiskers indicate significant differences compared to the control *E. coli* (Dunnett’s test, *n* = 9).

**Figure 6 microorganisms-09-00679-f006:**
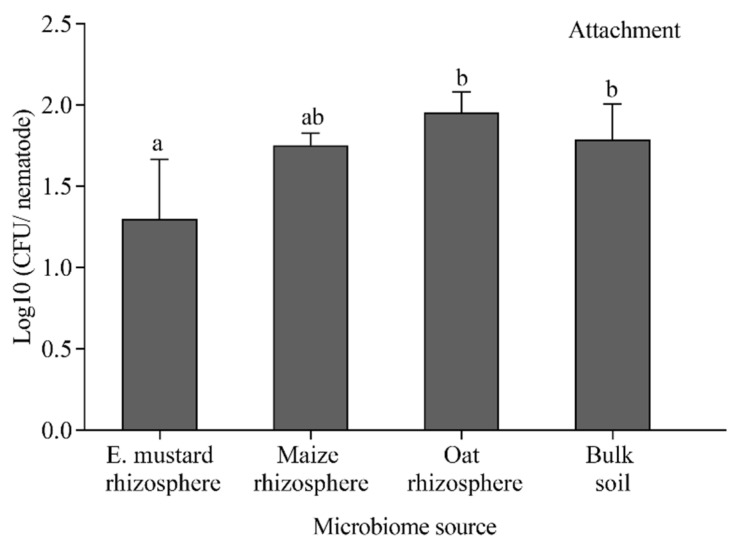
Number of microbes attached to infective stages of *Pratylenchus penetrans* after baiting in suspensions of soil from the rhizosphere of different crops or the corresponding bulk soil. Letters above bars indicate significant differences among treatments (Tukey’s test, *p* < 0.05, *n* = 12). Error bars represent standard deviations. E. mustard: Ethiopian mustard.

**Figure 7 microorganisms-09-00679-f007:**
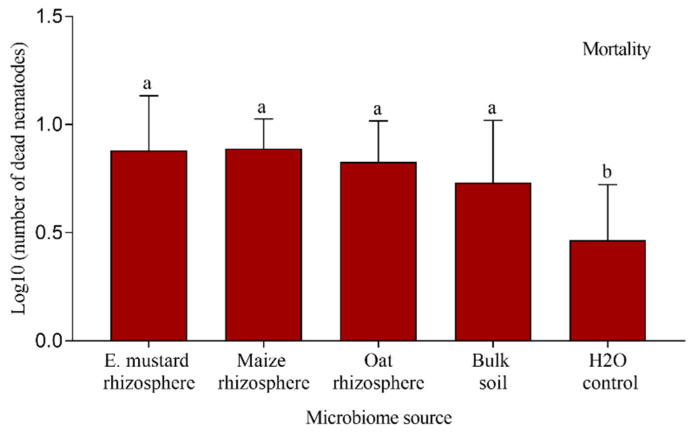
Effect of microbiomes from the rhizosphere of different crops or the corresponding bulk soil on mortality of *Pratylenchus penetrans*. Letters above bars indicate significant differences among treatments (Tukey’s test, *p* < 0.05, *n* = 12). Error bars represent standard deviations.

**Figure 8 microorganisms-09-00679-f008:**
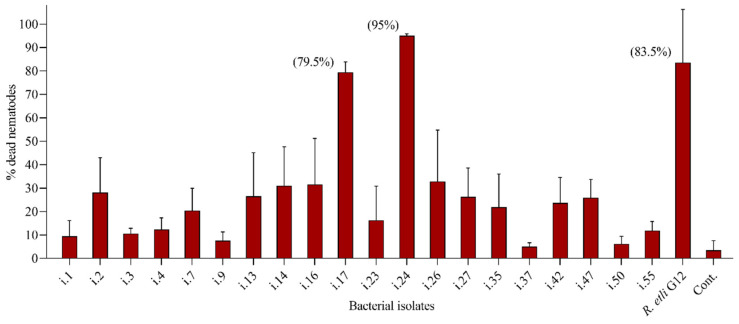
Effect of the nematode-isolated bacterial isolates on mortality of *Pratylenchus penetrans* after attachment to the cuticle. Cont.: negative control in sterile tap water. Error bars represent standard deviations, *n* = 3.

**Figure 9 microorganisms-09-00679-f009:**
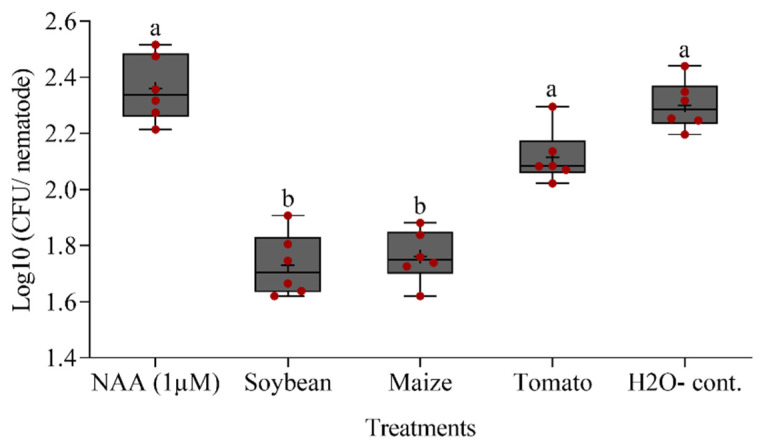
Effect of pre-incubation of *P. penetrans* in the root exudates of soybean, maize, or tomato on the attachment of the bacterial strain *Microbacterium* sp. i.14 to the surface of *P. penetrans*. Auxin (NAA) and sterile water were used as controls. Letters indicate significant differences among treatments (Tukey test, *p* < 0.05, *n* = 6). Medians are shown as (—), whiskers indicate quartiles.

**Table 1 microorganisms-09-00679-t001:** Taxonomic assignment of bacterial strains that were isolated from the cuticle of *Pratylenchus penetrans* after baiting in suspensions of bulk soil or different rhizosphere soils.

Isolate	Source	Most Similar Sequence, Accession No.	Identity
i.1	Bulk soil	*Streptomyces violaceoruber*, NR_112292.1	99%
i.3	Bulk soil	*Streptomyces atratus*, NR_043490.1	100%
i.4	Bulk soil	*Bacillus marisflavi*, NR_118437.1	99%
i.7	Bulk soil	*Nocardia coeliaca*, NR_104776.1	99%
i.9	Bulk soil	*Mycobacterium madagascariense*, NR_104690.1	99%
i.13	Bulk soil	*Microbacterium maritypicum*, NR_114986.1	99%
i.14	Bulk soil	*Microbacterium mangrovi*, NR_126283.1	99%
i.16	Bulk soil	*Delftia tsuruhatensis*, NR_113870.1	99%
i.17	Bulk soil	*Lysobacter capsici*, NR_044250.1	100%
i.23	Maize rhizosphere	*Novosphingobium aquaticum*, NR_148323.1	99%
i.24	Maize rhizosphere	*Bacillus cereus*, NR_074540.1	99%
i.26	Maize rhizosphere	*Pedobacter borealis*, NR_044381.1	99%
i.27	Maize rhizosphere	*Pseudomonas protegens*, NR_114749.1	99%
i.35	Soybean rhizosphere	*Bacillus megaterium*, NR_116873.1	100%
i.37	Soybean rhizosphere	*Alcaligenes faecalis*, NR_113606.1	99%
i.42	Soybean rhizosphere	*Rhizobium nepotum*, NR_117203.1	99%
i.47	Soybean rhizosphere	*Bacillus megaterium*, NR_116873.1	99%
i.50	Soybean rhizosphere	*Mycobacterium chubuense*, NR_041902.1	99%
i.55	Soybean rhizosphere	*Bacillus aryabhattai*, NR_115953.1,	100%
i.63	Tomato rhizosphere	*Staphylococcus capitis*, NR_113348.1	100%

**Table 2 microorganisms-09-00679-t002:** Percent dissimilarity of cuticle-attached bacterial or fungal communities after incubation of *P. penetrans* in suspensions of rhizosphere soils or bulk soil, based on denaturing gradient gelelectrophoresis (DGGE).

Sources of Soil Suspensions for Pairwise Comparison of DGGE Fingerprints of Nematode-Attached Bacteria or Fungi		Dissimilarity (%) ^a^	
		Bacteria	Fungi
**Experiment (1)**	Bulk soil vs. maize rhizosphere	23	62
	Bulk soil vs. tomato rhizosphere	35	88
	Bulk soil vs. soybean rhizosphere	40	65
	Maize vs. tomato rhizosphere	43	87
	Maize vs. soybean rhizosphere	51	43
	Soybean vs. tomato rhizosphere	68	57
**Experiment (2)**	Bulk soil vs. maize rhizosphere	34	14
	Bulk soil vs. oat rhizosphere	31	28
	Bulk soil vs. Ethiopian mustard rhizosphere	12	10
	Maize vs. oat rhizosphere	29	29
	Maize vs. Ethiopian mustard rhizosphere	36	17
	Oat vs. Ethiopian mustard rhizosphere	22	3

^a^ d-value: average of pairwise Pearson correlation coefficients among DGGE fingerprints within each group minus the average of pairwise Pearson correlation coefficients among DGGE fingerprints of different groups.

## Data Availability

All materials, isolates and clones culture stocks are available at Institute for Epidemiology and Pathogen Diagnostics, Julius Kühn Institute (JKI), Braunschweig, Germany upon request. DNA sequences of fungal ITS and bacterial 16S rRNA gene fragments were deposited in NCBI GenBank with accession numbers MN332046 to MN332063 and MW326933 to MW326970.

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
