# Peer review of "Plants Specifically Modulate the Microbiome of Root-Lesion Nematodes in the Rhizosphere, Affecting Their Fitness"

_microorganisms, 2021, doi:10.3390/microorganisms9040679_

Round 1
Reviewer 1 Report
It is a well-built manuscript with experiments and laboratory investigations requiring a lot of work. These results could help to understand the function of soil suppressivity against plant-parasitic nematodes better.
To begin with, I am not the expert of microbiology, I read the manuscript from the perspective of a nematologist.
Really helpful that the aim of the given experiment is briefly summarized at the beginning of the material and methods and also at the beginning of presenting results. This made the manuscript more understandable.
Theoretical question: You used HgCl2 in order to remove the old cuticule of P. penetrans individuals. What do you think, can the stress caused by this acid affect the outcome of the experiment?
Specific comments and editorial suggestions:
Ln 55: “the own microbiome of plants” instead of “the plant’s own microbiome”
Ln 71: “if” is not needed
Ln 73: expression “dialogue” is not clearly in this context. Please, rewrite this part.
Ln 86: “whether” instead of “if”
Ln 91: “using the” instead of “a” before “Baermann funnel”.
Ln 91: Does the reference “Elhady et al. [11]” belong to extraction as well, or only to surface sterilization? If it does and method you used is a modified version of Baermann funnel, indicate with “modified” before “Baermann funnel”, please. If does not, please refer to the original article.
Ln: 99-117: I don’t understand, what is the difference between Exp. 1 and Exp. 2. In both two experiments, soil was collected from two different fields and certain plants grown in it, right?
Ln 193: insert space between “1ml”
Ln 196-197: What type of microscope and on which magnification did you use?
Ln 411: The expression “surface disinfected nematode” appears in the first time. Use “surface-sterilized nematodes” consistently.
Ln 486 and 523: press enter
Ln 497: “in vitro” should be italic
As I understood, three main experiments were conducted (Ln 99: Microbiome associated with RLN as affected by plant species; Ln 171: Biological effects of nematode-attached microbiome; Ln 199: Effects of root exudates on microbial attachment to nematodes), for which supplementary laboratory investigations were linked. It could be better if these supplementary tests would be classified under each subchapter, for example: 2.2. Experimental design: microbiome associated with RLN as affected by plant species. 2.2.1. (instead of 2.3.) PCR-DGGE profiling of nematode-attached microbiomes.
Table 1.: In this table the point is, that you showed the difference between the bacterial communities of (e.g.) bulk soil and maize rhizosphere, which is 23%. Am I right? Would not been better to show exact values, like “bulk soil 10%, maize 33%”?
Recurrent comments:
Ln 66 (for example): I am aware that this manuscript is about root-lesion nematode Pratylenchus penetrans, and plant-parasitic nematodes in generally, but expressions should be more specific in some cases. For example, “nematode” could be bacterivore species, and I am not sure they could attack plants. So, this part could be more precise with “plant-parasitic or phytonematode attack”.
Ln 92 (for example): I don’t think that “sterile tap water” exists. Do you mean “sterilized tap water”? How did you sterilized tap (or any kind of) water?
Ln 118-119 (for example): “of nematode” instead of “nematode’s”
Ln 178 (for example): “g” of “5000 g” should be italic
Author Response
The suggested changes were done with Red
Theoretical question: You used HgCl2 in order to remove the old cuticle of P. penetrans individuals. What do you think, can the stress caused by this acid affect the outcome of the experiment?
HgCl2 was used not to remove the old cuticle, however, to surface-sterilized nematodes. Using HgCl2 to surface-sterilized nematodes is a common method for the in vitro and nematode microbe associations investigations
We agree with reviewer 1 and we are aware of the effect that can be caused by antibiotics and sterilization procedures. Therefore, we are using HgCl2 and antibiotics like the cell culture grad that used in the cell culture approach. As nematodes renewing their cuticle within few hours, we are always doing a step of incubation in sterile water for 48 hours before performing the experiments to allow the nematodes to renew and rebuild their cuticle followed by another washing step over sterile 5-µm sieves to wash out the debris of the old cuticle directly before the use in experiments.
Specific comments and editorial suggestions: and
Ln 55: “the own microbiome of plants” instead of “the plant’s own microbiome”
Done
Ln 71: “if” is not needed
Deleted
Ln 73: expression “dialogue” is not clearly in this context. Please, rewrite this part.
We will be appreciated if the reviewer can suggest how it should be rewritten.
Ln 86: “whether” instead of “if”
Done
Ln 91: “using the” instead of “a” before “Baermann funnel”.
Done
Ln 91: Does the reference “Elhady et al. [11]” belong to extraction as well, or only to surface sterilization? If it does and method you used is a modified version of Baermann funnel, indicate with “modified” before “Baermann funnel”, please. If does not, please refer to the original article.
The reference indicates only the sterilization protocol.
Ln: 99-117: I don’t understand, what is the difference between Exp. 1 and Exp. 2. In both two experiments, soil was collected from two different fields and certain plants grown in it, right?
Yes. To test this first hypothesis, we carried out two independent experiments with 2 different sets of plant species in two geographically different soil types to support and confirm our conclusion. In the first experiment, the rhizosphere microbiome was obtained from different plant hosts, maize, soybean, and tomato grown in field soil (less sandy loam with 1.4% humus, pH 6.2; 52°17'57" N, 10°26'14" E). In the second experiment, soil suspensions from the rhizospheres of maize, Ethiopian mustard, and oat plants are grown in field soil (sandy loam; 52°16'21.7"N 10°34'02.7"E) were obtained in the same way to study the attachment of bacteria and fungi to P. penetrans in this soil. We employed culture-independent (DGGE fingerprint, cloning to pick and identify the most abundant bacterial and fungal species attached to nematode cuticle which allows us in the near future to have a molecular marker to the most species associated with cuticle, we plasted the clones with the communities to explore which species relevant to which community). We also employed a culture-dependent method to isolate the culturable species associated with the cuticle for more studies in the future.
Ln 193: insert space between “1ml”
Done
Ln 196-197: What type of microscope and on which magnification did you use?
Done. The sentence of „using a stereomicroscope (Olympus Microscope SZX12)” was added.
Ln 411: The expression “surface disinfected nematode” appears in the first time. Use “surface-sterilized nematodes” consistently.
Done
Ln 486 and 523: press enter
Done
Ln 497: “in vitro” should be italic
Done
As I understood, three main experiments were conducted (Ln 99: Microbiome associated with RLN as affected by plant species; Ln 171: Biological effects of nematode-attached microbiome; Ln 199: Effects of root exudates on microbial attachment to nematodes), for which supplementary laboratory investigations were linked. It could be better if these supplementary tests would be classified under each subchapter, for example: 2.2. Experimental design: microbiome associated with RLN as affected by plant species. 2.2.1. (instead of 2.3.) PCR-DGGE profiling of nematode-attached microbiomes.
Good suggestion. We changed the order as you suggested.
Table 1.: In this table the point is, that you showed the difference between the bacterial communities of (e.g.) bulk soil and maize rhizosphere, which is 23%. Am I right? Would not been better to show exact values, like “bulk soil 10%, maize 33%”?
The pairwise Pearson correlation coefficients are based on the differences of the similarity between two groups or communities. Thus, it is interesting to show how different the community between two groups
Recurrent comments:
Ln 66 (for example): I am aware that this manuscript is about root-lesion nematode Pratylenchus penetrans, and plant-parasitic nematodes in generally, but expressions should be more specific in some cases. For example, “nematode” could be bacterivore species, and I am not sure they could attack plants. So, this part could be more precise with “plant-parasitic or phytonematode attack”.
Changed to “ against the attack of plant-parasitic nematode”
Ln 92 (for example): I don’t think that “sterile tap water” exists. Do you mean “sterilized tap water”? How did you sterilized tap (or any kind of) water?
Changed to sterilized. We sterilized the tap water by autoclaving
Ln 118-119 (for example): “of nematode” instead of “nematode’s”
Done
Ln 178 (for example): “g” of “5000 g” should be italic
Done

Reviewer 2 Report
Dear Authors,
I have an honor to revise paper entiled: “Plants specifically modulate the microbiome of root-lesion nematodes in the rhizosphere, affecting their fitness”. Authors presented interesting studies concepts concentrated on soil microbiome in the rhizosphere of different plant species, in two aspect - then they employed culture-independent and culture-dependent methods to study the microbial attachment to the cuticle of the nematode Pratylenchus penetrans.
First impression for the reader, that It is well designed and conceived the idea of proposed experiments.
Author revealed that Bacteria isolated from the cuticle belonged to Actinobacteria, Alphaproteobacteria, Betaproteobacteria, Gammaproteobacteria, Sphingobacteria, and Firmicutes. Moreover, interestingly, the isolates Bacillus cereus i.24 and L. capsici i.17 significantly antagonized P. penetrans after attachment.
Whereas, conditioning the cuticle of P. penetrans with root exudates significantly decreased the number of Microbacterium sp. i.14 attaching to the cuticle, furthermore, Authors postulated inducing changes of the cuticle structure. I suggest to analyze namatodes cuticle structure by scanning electron microscope to confirm this kind of hypothesis.
It’s commonly known fact, that microbial community architecture/structure in the rhizosphere was highly dependent on the plant species, but please explain, why exactly these three plants were taken into account to analyze rhizosphere interactions -mustard, maize, and oat.
I have also a question, did isolation procedure of bacteria from nematodes cuticle have not influenced on cuticle structure, like in point 2.4 M&M ?
When Author analysed root exudates investigate different rhizosphere influence – roots from soybean, maize and tomato, please explain why?
It will be very usefull and “sound wider” to add future prospects coming from authors promising/interesting findings in conclusion chapter.
Author Response
Whereas, conditioning the cuticle of P. penetrans with root exudates significantly decreased the number of Microbacterium sp. i.14 attaching to the cuticle, furthermore, Authors postulated inducing changes of the cuticle structure. I suggest to analyze namatodes cuticle structure by scanning electron microscope to confirm this kind of hypothesis.
It is a good suggestion to analyze cuticle structure in response to attached microbes as affected by environmental factors in the next step in a follow-up study. We are aware that our approach and the results have the potential to trigger further studies. The scanning electron microscope or the atomic microscope are good tools for such studies. We hope that this publication will trigger such efforts
It’s commonly known fact, that microbial community architecture/structure in the rhizosphere was highly dependent on the plant species, but please explain, why exactly these three plants were taken into account to analyze rhizosphere interactions -mustard, maize, and oat.
We tested the first hypothesis in two independent experiments with 2 different sets of plant species in two geographically different soil types to support and confirm our conclusion.
In the first experiment, the rhizosphere microbiome was obtained from different plant hosts, maize, soybean, and tomato, in the second experiment maize, Ethiopian mustard and oat.
Those plant species were used as those plants are used as possible rotated crops with soybean under temperate condition in Europe. We are currently investigating the effect of microbiomes due to plant-soil feedback of those plant species on the nematode suppression in soybean. The microbiome associated with nematodes incubated in the rhizosphere of those crops will help us to know which microbes are enriched in the rhizosphere of those crops and potential attached to the nematode cuticle and understand how the nematode then are suppressed.
I have also a question, did isolation procedure of bacteria from nematodes cuticle have not influenced on cuticle structure, like in point 2.4 M&M ?
we are aware of the effect that can be caused by antibiotics and sterilization procedures. Therefore, we are using HgCl2 and antibiotics like the cell culture grad that used in the cell culture approach. As nematodes renewing their cuticle within few hours, we are always doing a step of incubation in sterile water for 48 hours before performing of the experiments to allow the nematodes to renew and rebuild their cuticle followed by another washing step over sterile 5-µm sieves to wash out the debris of the old cuticle directly before the use in experiments.
When Author analysed root exudates investigate different rhizosphere influence – roots from soybean, maize and tomato, please explain why?
We selected the root exudates of these crops to study this particular hypothesis. NAA was found in several studies to be important in triggering changes in the surface lipophilicity of the surface of the plant-parasitic nematode and mimic phytohormones diffused from roots.
It will be very usefull and “sound wider” to add future prospects coming from authors promising/interesting findings in conclusion chapter.
The plant-soil feedback theory is a future perspective of this study and it is a current-running project where we are trying to find out how we can harness the rhizosphere microbiome from the previous crop to suppress the nematode in the coming or the follow-up crop. We wish that this publication will trigger such efforts.